# You Only Look at One:
# Category-Level Object Representations
# for Pose Estimation From a Single Example

**Walter Goodwin, Ioannis Havoutis, Ingmar Posner**
Oxford Robotics Institute
University of Oxford
`firstname@robots.ox.ac.uk`

**Abstract:** In order to meaningfully interact with the world, robot manipulators must be able to interpret objects they encounter. A critical aspect of this interpretation is pose estimation: inferring quantities that describe the position and orientation of an object in 3D space. Most existing approaches to pose estimation make limiting assumptions, often working only for specific, known object instances, or at best generalising to an object category using large pose-labelled datasets. In this work, we present a method for achieving category-level pose estimation by inspection of just a single object from a desired category. We show that we can subsequently perform accurate pose estimation for unseen objects from an inspected category, and considerably outperform prior work by exploiting multi-view correspondences. We demonstrate that our method runs in real-time, enabling a robot manipulator equipped with an RGBD sensor to perform online 6D pose estimation for novel objects. Finally, we showcase our method in a continual learning setting, with a robot able to determine whether objects belong to known categories, and if not, use active perception to produce a one-shot category representation for subsequent pose estimation.

## 1 Introduction

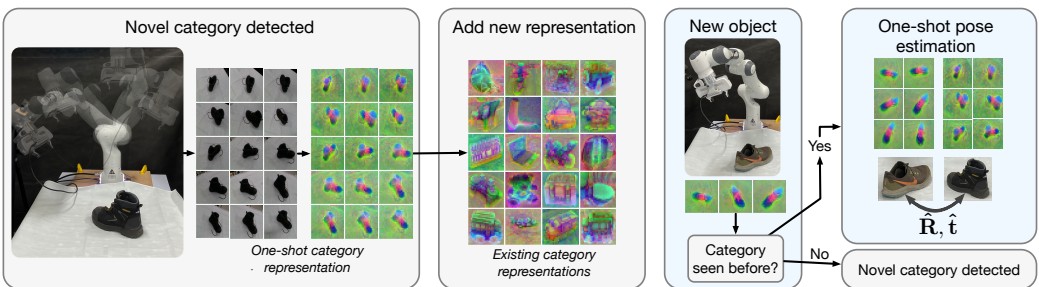

**Figure 1:** An overview of our method. Multiple views of an object from a new categories are taken and turned into a category-level representation in a 10 s process. When a novel object is encountered, it is compared to all current representations. If it is deemed to belong to a previously captured category, our method performs 6D pose estimation in real-time. If it is not, it is scanned and a new category-level representation acquired. The approach runs in real-time and representations allow accurate pose estimation of unseen objects from the captured categories.

Many practical applications of robotics require that a robot is able to estimate the poses of objects that are encountered, that is, the parameters that describe their position and orientation in 3D space relative to some coordinate frame. This capability is central to object manipulation, where to successfully rearrange objects, a robot needs to estimate object pose to infer how it differs from some

6th Conference on Robot Learning (CoRL 2022), Auckland, New Zealand.

goal state [1]. Object pose estimation can also be helpful for mobile robots seeking to localise themselves in the world, by providing semantically grounded anchors [2, 3].

Despite its central importance to robotics, approaches for object pose estimation remain limited in their abilities to generalise to novel objects. State of the art approaches often demand large labelled datasets or accurate CAD models just to produce pose estimators for single object instances [4, 5]. Over the last few years, attempts have been made to alleviate these issues by learning *category level* pose estimators that train over multiple object *instances* from a category and can generalise to unseen instances from that category [6, 7, 8]. Such generalisation to unseen objects is a much sought-after property in robotics, with some key examples being generalised manipulation [9, 10, 11] and object-based SLAM [2]. However, these category-level approaches to pose estimation still have substantial data requirements in order to generalise - for instance, the Objectron dataset [12, 8] has over 17,000 object instances and 4 million frames to cover 9 object categories.

Separately, some recent works have achieved very promising results on pose estimation for *single* object instances by avoiding object-specific deep models, instead collecting a set of reference views of an object before performing robust feature matching for view retrieval and pose estimation [13, 14, 15]. These approaches remove the need for large curated datasets required by deep learning approaches, but are designed for single object instances and thus fail to exhibit desirable category-level generalisation.

In this work, we present an approach to object pose estimation that has the desirable properties of both of these paradigms. By leveraging the pre-trained features of a deep vision transformer (ViT) network [16], we show that we are able to leverage multiple views of a reference object to achieve pose estimation on *any novel object from the same category*. Extending recent work on leveraging such features for pose estimation [17], we demonstrate much improved performance by also making use of multiple views of the novel target object. Further, in contrast to prior work, both the processing of object reference views into a reference object 'model' suitable for downstream pose estimation, and the pose estimation process itself, can run in real-time and are thus suitable for in-the-wild robotic deployment. We demonstrate that our method drastically outperforms alternative methods in this one-shot category-level pose estimation setting, and show that this approach enables a robot manipulator to perform generalised manipulation tasks, in which a single 'goal' pose for an object category is inspected by a robot manipulator, which then performs pick-and-place rearrangement to satisfy this goal pose when confronted with novel items from the same category.

In summary, our main contributions are:

- A training-free, multi-view approach to object pose estimation that only requires views of a single reference object, yet generalises to novel objects from the same category.

- A fast implementation of our pose estimation method that runs orders of magnitude faster than the closest baseline, enabling both real-time capturing of objects from novel categories, and real-time pose estimation for objects from known categories.

- Validation of the method's utility in a continual learning setting, in which the method is deployed tabula rasa and distinguishes novel categories from known categories, learning representations where appropriate.

## 2 Related Work

### 2.1 Category-level object pose estimation

Approaches to category-level object pose estimation can generally be divided into those which learn a category-level representation against which novel objects and images can be compared [6, 18, 19, 20, 21, 22], those which leverage (a set of) CAD models for a category [23, 24, 25, 26], and those which train deep models to directly regress to parameters describing object pose [27, 28, 29]. These methods require large pose-labelled datasets containing multiple instances of objects from each category of interest in order to generalise successfully.

In contrast, in this work we build on a recent paper that proposes a novel zero-shot category-level pose estimation setting [17], in which, given a single image of some reference object and no pose-labelled data, the task is to estimate the pose of novel objects from the same category, using a handful RGB-D views. Building on [17], our work enables the use of multiple reference object views, and we show that this leads to a considerable improvement in performance. Further, while [17] takes over 5 seconds to estimate pose for a single frame, our method can be run at over $15\,\mathrm{Hz}$, enabling real-time pose estimation on a robot manipulator.

## 2.2 Template-based object pose estimation

In making use of a collection of reference views of an object for pose estimation, our work is related to the literature on template matching. Template matching for pose estimation entails collecting many views of a reference object, and retrieving the closest view at run-time through a visual similarity measure. Recent work achieves impressive performance for single instances by constructing a point cloud of an object, and leveraging learnt 2D-to-3D correspondence matching and Perspective-n-Point (PnP) to recover pose when the object is next encountered [30]. This method just requires an object point cloud to perform pose estimation for novel objects. Similarly, [15] uses a two-step matching process with dense CNN features, before a final PnP step is used to estimate object pose. Similar viewpoint-matching approaches have been used for pose tracking during manipulation [14]. [31] solves a continual learning case, in which 3D object representations are captured when an object is first encountered by a robot, and subsequently matched against in a render-and-compare manner to provide 6D pose estimation. Template matching approaches have recently been employed in a few-shot setting, where following a pre-training stage on a diverse dataset, just a few views of a novel object are sufficient to estimate accurate pose [32]. All of the aforementioned methods enable pose estimation of novel objects provided that a collection of reference views, or a 3D model, can be acquired. However, the matching processes they employ are instance-specific, meaning that object templates cannot be used for pose estimation of previously unseen objects from related categories.

## 2.3 Category-level object representations for manipulation

Several prior works seek to learn category-level object descriptors to enable generalising robot manipulation skills from single object instances to broader object categories. [9] learn dense object descriptors that can be trained in a mostly self-supervised manner from multiple views, and show that for several categories, after training on  15 object instances nearest-neighbours in descriptor space between different objects tend to correlate with meaningful correspondences. Other work has sought to learn category-level *keypoints* [10]. Manipulation goals described as costs on keypoint positions can then be achieved on novel objects from known categories. More recently, category-level descriptors have been learnt on 3D object representations [11], by self-supervised training over a large set of CAD models from a category. These descriptors enable finding oriented correspondences between semantically equivalent points on objects from a training category, and have been used to place novel objects in target poses with a robot manipulator. These prior works require time-consuming training, over datasets collected from a not inconsiderable diversity of objects from a category of interest, and thus do not scale well. In this work, we show that dense object descriptors that are similarly able to produce meaningful part-based correspondences between objects from a given category can be achieved through inspection of just a single object instance in real-time. As in prior work, these descriptors can be used to match object poses, but in a considerably more scalable manner, with minimal data requirements.

# 3 Methods

## 3.1 One-shot pose estimation setting

To formalise the setting for one-shot pose estimation addressed in this work, we adopt and extend the notation from [17]. We consider an object from a category $c$ to be represented by $M$ views, to

produce a reference image set, $I_{\mathcal{R}_{1:M}}$. Except in certain ablations, we assume that depth images are also available from these views, $D_{\mathcal{R}_{1:M}}$. We assume that camera extrinsics are known, a realistic assumption in most robotics contexts (e.g. a manipulator with a wrist-mounted camera), and denote these as $\mathbb{T}_{\mathcal{R}_{1:M}}$ for the reference object views. At pose estimation time, a target object is encountered, and we assume that $N > 1$ views of this object are also captured, with RGB images $I_{\mathcal{T}_{1:N}}$, depth images $D_{\mathcal{T}_{1:N}}$, and camera extrinsics $\mathbb{T}_{\mathcal{R}_{1:N}}$. We seek to recover a rotation $\hat{\mathbf{R}}$ and translation $\hat{\mathbf{t}}$ that describes the pose of the target object with respect to the reference object.

The reference object in the one-shot pose setting serves both as a model against which target object correspondences for pose estimation are made, but also to establish a canonical frame against which target object pose is measured. Previous work describes a similar setting in which just a single reference view is used as entailing 'zero-shot' pose estimation, because this reference view is a bare minimum in order to render the pose-estimation problem well-posed. In our setting, because multiple reference views of an object are aggregated to form a representation for a category, we choose to describe this as one-shot pose estimation. Estimating the pose of a novel object *relative* to the reference object is sufficient to generalise manipulation behaviours demonstrated on the reference object.

## 3.2 Descriptors and correspondences

[17] uses features extracted from a vision transformer network (ViT) for category-level pose estimation, showing that they generalise well across instances within a category. In this work we use the same ViT, trained with DINO [16], an unsupervised contrastive method that uses a multi-scale cropping process during training which, intuitively, encourages the network to discover correspondences between small local crops of an image, and a larger global crop. Empirically, DINO's features can identify objects but also object *parts*. Further, ViTs take a patch-based approach to image processing which retains relatively high spatial resolution throughout the network, in contrast to CNNs, which spatially downsample such that later layers with expressive features lack spatial resolution. By using a ViT with a patch size of 8x8 pixels on RGB images resized to 224x224, the resulting feature maps at all layers are 28x28. This enables patch-based correspondences to be localised accurately in the image, and thus in 3D space through backprojection. We denote normalised feature maps extracted from image $I_i$ as $\Phi(I_i) \in \mathbb{R}^{28 \times 28 \times D}$, where D is the descriptor dimensionality (see Section 3.3).

We construct correspondences between the reference and target objects by finding $P$ patch correspondences between every reference-target feature map pair $\{\Phi(I_{\mathcal{R}_i}), \Phi(I_{\mathcal{T}_j})\}_{i \in 1:M}^{j \in 1:N}$. For an image pair $I_{\mathcal{R}_i}, I_{\mathcal{T}_j}$ we denote these $\{u_{i_k}, v_{j_k}\}_{k \in 1:P}$, where $u$, $v$ are indexes into the respective feature maps. The problem of finding strong correspondences between two feature maps $\Phi_1$, $\Phi_2$ is much studied, with many proposed solutions. Naive nearest neighbours can result in many-to-one matches that are physically implausible and thus not appropriate for downstream pose estimation. Mutual nearest neighbours [33] is a stronger condition, in which points $u$, $v$ are considered correspondences only if the closest descriptor to a point $u$ in $\Phi_1$ is $v$ (the standard nearest neighbours condition) *and* the closest descriptor to $v$ in $\Phi_2$ is $u$. Formally, for $v = \text{argmax}_w d(\Phi_{1_u}, \Phi_{2_w})$, we have $u' = \text{argmax}_w d(\Phi_{2_v}, \Phi_{1_w}) = u$, where $d(\cdot, \cdot)$ is the cosine similarity. Requiring this perfect 'cycle' between the descriptors reduces spurious correspondences, but cannot guarantee that a certain number - or indeed any - correspondences will be found, which affects downstream pose estimation in our method.

To ensure that $P$ correspondences are found for each reference-target pair, we use the cyclical distance metric proposed in [17], which can be thought of as a relaxation of mutual nearest neighbours matching: if there are $< P$ mutual nearest neighbours, additional correspondences are given points for which $u' - u$ as defined above is minimal. This acts as a spatial prior: correspondences that are *almost* spatially consistent under the cycle are preferred: $u'$ and $u$ are likely to still belong to the same semantic *part* of an object. We also experiment with a novel variant of this approach, in which the 'cyclical distance' is measured in 3D world coordinates rather than pixel space, using the back-projected pixel coordinates (details in supplementary).

## 3.3 Dimensionality reduction

In well-established approaches to learning category-level dense descriptors for robotic manipulation (e.g. [9]), descriptor dimensionality tends to be low. The authors of [9] note that for single instances, a 3-dimensional descriptor space tends to be sufficient to discriminate between object parts, though this requirement scales somewhat for multi-object descriptor networks. The raw ViT features we use as descriptors in this work are 384-dimensional. Such dimensionality is critical to the network's ability to satisfy the contrastive objective over the diverse dataset it is trained on. At the pose estimation stage, we would like to retain those factors of descriptor variation that enable us to distinguish different object parts and viewpoints, but it is natural to wonder whether there may be considerable redundancy in this descriptor space. To explore this, we perform PCA on the descriptors taken over all reference object views. The majority of descriptor variance is captured in a relatively small number of descriptor dimensions (supplementary). Further, we find that projecting features onto a small subset of principal components actually results in *improved* pose estimation results. Certain results are show in Table 2 and discussed in Section 4.1.1. The category representations in Fig. 1 are the projections of descriptors onto the first three principal components.

In our experiments, we calculate principal components based on the *reference* feature maps only, and use these to project both reference and target descriptors prior to finding correspondences.

## 3.4 Viewpoint estimation

### 3.4.1 Aggregating multi-view correspondences

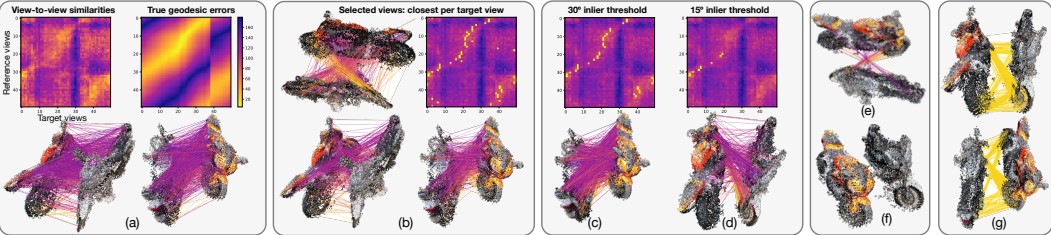

**Figure 2:** For visualisation, point clouds are created by a threshold on ViT attention masks to mask each view, aggregating, and removing outliers. **(a)** The matrices show (*left*) the view-to-view similarities (sum of top-K correspondences) for each reference-target view pair ( ▢ =high, ▪ =low), (*right*) the true geodesic error between the object poses in these views, which can be seen to be correlate with the similarities. Below, a random subset of correspondences across all view pairs is visualised. **(b)** For each target view, the corresponding reference view with the highest similarity is chosen, with these pairs highlighted in yellow on the matrix. A subset of the resulting filtered correspondences is shown. **(c)** The resulting inlier frames by consensus filtering with a threshold of $30°$ (Section 3.4.1), and correspondences. **(d)** The same with a tighter inlier threshold of $15°$. **(e)** The inlier correspondence set following RANSAC. **(f)** Objects aligned by estimated pose, offset by a translation for clarity. **(g)** Taking top-K correspondences globally from the whole matching process (rather than filtering to good views) leads to conflicting and erroneous correspondences.

Following [17], we assign similarities $\mathcal{S}_{ij}$ to reference-target view pairs e.g. $I_{\mathcal{R}_i}, I_{\mathcal{T}_j}$ as the sum of the similarities between the top-K correspondences for this pair, $\{u_{i_k}, v_{j_k}\}_{k \in 1:P}$, found as described previously. That is, $\mathcal{S}_{ij} = \sum_{k=1}^{P} d(u_{i_k}, v_{j_k})$. Considering all reference images $I_{\mathcal{R}_{i=1:M}}$ and target images $I_{\mathcal{T}_{j=1:N}}$, we arrive at a view similarity matrix $\mathbb{S}_{M \times N}$, with elements $\mathbb{S}_{M \times N}[i, j] = \mathcal{S}_{ij}$. An example of such a similarity matrix, and subsequent processing steps and their effect on the correspondence set, is shown in Fig. 2. We leverage this matrix to filter to a subset of best-fit view pairs. First, for each target frame we retain just the reference frame with maximum similarity. Subsequently, we further filter based on the estimated relative orientation estimates we would arrive at if we took each pair to represent a *perfect* alignment. With this set of relative poses, we find the largest consensus set by sampling many random poses in SO(3) and finding the viewpoint pairs with a relative pose within some threshold $\theta°$ distance from this pose. These largest such set is chosen. Correspondences from these remaining views are used in the final rigid body transform solution.

### 3.4.2 Final pose estimation

Given a filtered subset of reference-target view pairs which have a consensus on an approximate SO(3) pose prediction, we seek to estimate a refined SO(3) pose, along with a translation, to produce a full 6D object pose estimate, and a scaling, to handle intra-category size differences. For this, we assume that the reference and target objects are related by such a 7-D rigid-body transform, and solve for this using RANSAC and Umeyama's method [34]. With $K$ correspondences per view pair and a subset of $Q$ view pairs, we have correspondences $\{u_k, v_k\}_{k=1:QK}$. Using reference and target depth maps $D_{\mathcal{R}_{i=1:M}}$ and $I_{\mathcal{T}_{j=1:N}}$, we back project these pixel coordinates to 3D points $\{\mathbf{u}_k, \mathbf{v}_k\}_{k=1:QK}$. For each RANSAC trial we sample four pairs of corresponding points as required by Umeyama's method, and the estimated rotation $\hat{\mathbf{R}}$, translation $\hat{\mathbf{t}}$ and scaling $\hat{\lambda}$ satisfy

$$(\hat{\lambda}, \hat{\mathbf{R}}, \hat{\mathbf{t}}) = \operatorname*{argmin}_{(\lambda, \mathbf{R}, \mathbf{t})} \sum_{k=1}^{4} \mathbf{v}_k - (\lambda \mathbf{R} \mathbf{u}_k + \mathbf{t}) \qquad (1)$$

We run RANSAC for 1000 trials and make a final estimate using the largest resulting inlier set.

### 3.5 Continual category learning

An often overlooked upstream requirement of most object pose estimation methods used in robotics is object detection and classification. Many approaches to category-level *and* instance-level pose estimation require knowledge of the target object's category or identity to choose the appropriate network [8], network head [35], or template [30]. We would like to be able to leverage the one-shot setting presented in this work in a fully autonomous context, where the category identities of objects encountered by a robot are estimated for subsequent pose estimation. Specifically, a compelling setting in robotics is that of continual learning, where a robot encountering novel objects can either assign them to categories seen before, or determine that they are a novel category. In our case, the former would lead to a selection of a suitable reference object for pose estimation, while in the latter, the robot would employ active perception to produce multiple views of this novel object, forming a representation for a novel category. We denote the set of previously 'discovered' categories $\mathcal{C}_{1:S}$, and given one or more views of a novel object $I_{\mathcal{T}_{1:N}}$, wish to determine the object's identity $\hat{\mathcal{C}} \in \{C_1...C_{S+1}\}$, where $C_{S+1}$ would imply this object belongs to an unseen category. We seek a similarity metric that enables fast checking against a potentially large number of reference objects. For this, we use the [CLS] tokens from the last layer of the DINO ViT [16]. For an existing category, we represent it as the set of [CLS] tokens over all sequences and views that have been assigned to it, such that the whole feature set is $\{\Psi_s = \psi_{s_1}, ..., \psi_{s_F}\}_{s=1:S}$. For a new category with [CLS] tokens over its views $\psi_{x_1}, ..., \psi_{x_G}$, we assign the category $s$ as:

$$\operatorname*{argmax}_{s} f(\Psi_s, \Psi_x) \text{ if } f(\Psi_s, \Psi_x) > \theta, \text{ where } f(\Psi_s, \Psi_x) = \Big( \sum_{i=1}^{F} \sum_{j=1}^{G} d(\phi_{s_j}, \phi_{x_i}) \Big) \qquad (2)$$

Where $\theta$ is a threshold on similarity (details in supplementary). If no category meets this threshold, the object is assumed to belong to a novel category.

## 4 Experimental Results

### 4.1 Multi-view one-shot pose estimation

To evaluate the performance of the proposed pose estimation method on a diverse range of categories and instances, we use a pose-labelled subset [17] of the Common Objects in 3D (CO3D) dataset [36], for which the ground truth relative pose between video sequences of 10 distinct objects from each of 20 categories is labelled. Each sequence contains approximately 100 frames, and is taken by a smartphone in a hand-held turntable-style object scan. Camera extrinsics and depth images in this dataset are approximate, being recovered from RGB video sequences by a Structure-from-Motion approach [37]. While we find that (with appropriate real-time depth completion, described

| | All Categories | | | | Per Category (Acc@30°), % | | | | | |
|---|---|---|---|---|---|---|---|---|---|---|
| Method | Med. Err (↓) | Acc30° (↑) | Acc15° | Acc7.5° | B'pack | Car | Chair | Keyboard | Laptop | M'cycle |
| TEASER++ [13] | 126.4 | 3.75 | 1.1 | 0.3 | 1 | 5 | 9 | 8 | 6 | 1 |
| Goodwin2022 [17] | 47.7 | 49.4 | 28.35 | 9.6 | 44 | 65 | 47 | 69 | 85 | 85 |
| Ours-R | 58.0 | 29.9 | 13.4 | 4.4 | 44 | 23 | 34 | 15 | 63 | 73 |
| Ours-RC | 32.7 | 56.6 | 35.0 | 13.1 | 64 | 74 | 58 | 57 | 91 | 82 |
| Ours-U | 36.9 | 59.4 | 45.7 | 26.6 | 49 | **92** | 59 | **80** | 97 | **100** |
| Ours-UC | 31.9 | 61.5 | 47.5 | 27.4 | 61 | 78 | 64 | 67 | 98 | 99 |
| Ours-UCD | 26.2 | 64.5 | 49.4 | 29.2 | 60 | 87 | 70 | 70 | 99 | 99 |
| Ours-UCD+ | **21.4** | **69.8** | **54.6** | **34.5** | **67** | 86 | **76** | 76 | **100** | **100** |

**Table 1:** Pose estimation accuracy (orientation). We report Acc@$\theta°$, with $\theta \in \{30°, 15°, 7.5°\}$, giving the percentage of estimates that fall a certain maximum threshold on geodesic error, and the median error 'Med. Err' (per category, then averaged). We compare to TEASER++ [13], and Goodwin2022 [17]. Suffixes on our method ablations: **R**: retrieval; pose estimated based on mean of most similar views. **C**: consensus, iterative removing far-from-mean views. **U**: Umeyama's method; rigid body solution using best-view correspondences. **D**: descriptor dimensionality reduction (to 32 components). Methods use 10 reference and target views (Goodwin2022 uses just 1 reference). **+** indicates 30 views used.

in supplementary) the extrinsics and depth are of sufficiently high quality, we consider this dataset to be a lower bound on data quality from a wrist-mounted RGB-D camera.

The results of our method and certain ablations are shown in Table 1. We compare to the method of [17], which uses a single reference image, and to TEASER++, a fast point cloud registration algorithm that works in the presence of a large number of outliers, unknown correspondences, and - unlike ICP - does not require a strong initial guess. Further, it leverages an explicit bound on the noise it assumes present in the *non-outlier* points, which is analogous to the inlier threshold we use in our RANSAC stage (see supplementary), and thus forms an interesting baseline for leveraging *purely* geometric correspondences for category-level pose estimation.

The improvement over [17] can be seen especially strongly in at the more precise Acc@15° and Acc@7.5°, where our full method demonstrates almost double and triple the accuracy respectively, and scores 20.4% higher on Acc@30°. This underlines the importance of aggregating many-to-many view comparisons to arrive at pose estimates. While [13] shows better-than-random performance for most categories, it is not in general able to solve the category-level pose estimation problem, indicating that stronger priors than just geometric distances are necessary in this setting. Further results, including those for translation errors, are in the supplementary.

### 4.1.1 Effect of descriptor dimensionality reduction

In Section 3.3, we discuss our approach to reducing descriptor dimensionality. Table 2 shows the impact of using differing numbers of principal components on the performance of our pose estimation method. We find that projecting descriptors onto a small subset of principal components not only serves to reduce both computation time, and memory usage per reference object, but also have a beneficial effect on performance. For 10, 20 and 30 view regimes, we see 8%, 2% and 3% improvements respectively when reducing from 384D to 32D descriptors. An intuitive explanation for the power of retaining just the top principal components from multi-view features is that doing so discards directions in feature space which are *not* variant under viewpoint. For instance, a ViT feature for a patch containing a wing-mirror of a red motorbike might capture the semantic nature of this part - which is important spatially for predicting pose - but might also capture the redness: the latter is invariant over the views, so would not feature in the first principal components. This would be an example of an instance-specific property that might be encoded in the full feature dimensionality, yet would be detrimental to finding strong cross-instance correspondences.

### 4.2 Novel category discovery

Having verified that our method can produce accurate pose estimation for a range of objects using just a single reference object for each category, we note that in the setting above, we assume that the object identities are known. That is, we never attempt to predict a bicycle's pose from a teddy-

| Views | # Dim | Med. Err | Acc@30° | Acc@15° | Acc@7.5° | Time (ms) | Memory (MB) |
|---|---|---|---|---|---|---|---|
| 10 | 384 | 25.8 | 61.4 | 39.7 | 17.2 | 2.96 | 12.0 |
| 10 | 32 | 23.7 | 66.2 | 43.0 | 18.3 | 2.60 | 1.0 |
| 10 | 6 | 32.0 | 51.6 | 29.5 | 11.2 | 2.60 | 0.2 |
| 10 | 3 | 59.2 | 35.8 | 17.1 | 5.9 | 2.60 | 0.1 |
| 20 | 384 | 22.1 | 68.4 | 49.6 | 25.9 | 4.06 | 24.1 |
| 20 | 32 | 22.0 | 69.8 | 50.1 | 24.0 | 3.36 | 2.0 |
| 30 | 384 | 20.4 | 69.9 | 51.4 | 25.4 | 5.18 | 36.1 |
| 30 | 32 | **20.3** | **72.1** | **52.4** | **26.2** | 4.23 | 3.0 |

**Table 2:** Pose estimation (orientation) performance over 20 object categories, with varying descriptor dimensionality reduction and number of reference and target views. Dimensionality reduction by an order of magnitude (from 384D to 32D or 24D) both reduces computational burden *and* improves results.

bear reference. Our approach for continual learning of categories (Section 3.5) should enable such identity matching (classification and novel category discovery) to be fully autonomous. To assess this, we simulate 1,000 continual learning episodes using 10 sequences from each of 20 CO3D categories. In each episode, 250 sequences are drawn sequentially and at random. For each sequence we either assign it to an existing or a novel category as described in Section 3.5. Objects that are erroneously believed to be novel categories are False Negatives (FN), and those assigned to a set where they are not the majority true category are False Positives (FP). Through tuning the similarity threshold, a 0% FP rate can be reached with a 26% FN rate.

### 4.3 Robotic novel object pose rearrangement

We deploy our method on RealSense D435i camera wrist-mounted on a Panda robot. To scan objects from novel categories, we sample 10 or 20 views in an approximate hemisphere above the object, with the camera oriented towards the object. When an object is detected as belonging to a known class (Section 3.5), the robot traverses between several random waypoints that look at the object, and we inspect pose estimates through a visualised oriented 3D bounding box. Further details are in the supplementary.

## 5 Limitations

Our work assumes that camera extrinsics are available between frames at runtime, and for best performance also assumes that a depth camera is available. While these are both readily satisfied in the motivating example of a robot manipulator with a wrist-mounted camera, these conditions may not be so readily available in, for example, an AR context in which a moving mobile phone is the candidate device for running pose estimation. However, the correspondence component of our method can still be used in a depth-free setting (see Section 4.1), with either Perspective-n-Point if depth is still available for the *reference* object, or essential matrix estimation if depth is available for neither. This work assumes that the relationship between different objects within a category can be captured reasonably well by a 7-parameter rigid-body transform between the spatial locations of corresponding salient points on the objects. This would not hold for deformable or articulated objects, or objects categories exhibiting drastic shape diversity. Relaxing this assumption would be a promising direction for future work. An extension to the proposed method which models inter-instance relationships with a 9-D affine transform is described in the supplementary.

## 6 Conclusion

In this work we propose an approach to object pose estimation that runs in real-time on RGB-D data, and can provide accurate pose estimates at the category level by leveraging multiple views of a single reference object from the category. We show that high dimensional features from a pre-trained ViT can become high quality and efficient descriptors through dimensionality reduction. We demonstrate that our method can function as a fully autonomous approach to continual learning of category representations for pose estimation, with a scalable approach to classification of novel objects. Our method runs at over 15 Hz on a camera wrist-mounted on a robot arm.

**Acknowledgments**

The authors gratefully acknowledge the use of the University of Oxford Advanced Research Computing (ARC) facility http://dx.doi.org/10.5281/zenodo.22558. We thank Jack Collins, Oiwi Parker Jones, and Sagar Vaze for looking at early drafts.

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
