# OpenReview forum: "You Only Look at One: Category-Level Object Representations for Pose Estimation From a Single Example"
_robot-learning.org/CoRL/2022/Conference — CoRL 2022 Poster_

### Official Review · Reviewer_5Bj6 · 2022-07-17

**Originality:** Very Good
**Technical Quality:** Excellent
**Clarity Of Presentation:** Excellent
**Impact:** 4

**Recommendation:**

Strong Accept: I recommend accepting the paper and will argue for my recommendation even if other reviewers hold a different opinion.

**Summary:**

The paper presents a method for category-level object pose estimation based on RGB-D input. The approach does not require pose-labeled data, 3-D models, or category-specific networks, but only relies on few views of a target object and of a reference one of the same category. Similarly to [17], the paper shows that features extracted from a pre-trained ViT can be used as descriptors to find category-level correspondences for pose estimation. Compared to [17], the paper proposes to use multiple views from the reference object and shows that this has a significant impact on the estimated poses. Further, the method is applied in a continual learning setting for novel category discovery. The evaluations show that proposed approach significantly outperforms [17] and the baseline of [13], which however uses purely geometric correspondences. The method is demonstrated in a robotic setup to perform real-time pose estimation of a novel object based on a reference object from the same category.

**Issues:**

### Technical details
- L144: In what sense "normalised"?
- Sec. 3: What values were used for $M$ and $N$? Was an ablation study performed to select them?
- Fig. 2: "Three consensus filters"/"iterations of filtering": Where does the "three" come from? Does this value always guarantee that the criteria mentioned in the text (L197) are met?
- Fig. 2: "good" -> "closest", as in the subfigure title of (b)? That is, what does "good" mean here?
- Sec. 3.4: What value of $K$ was used? What impact does this have on the resulting pose estimate?
- Sec. 3.4: The correspondence indices $u_k, v_k$ come from a $28\times 28$ feature map, whereas the RGB-D image has larger resolution. How do you select the depth of the 3-D point associated to $u_k, v_k$, given the difference in resolution? Have you investigated matching the RGB-D resolution, e.g., using interpolation? Intuitively, this could produce more accurate estimates of the 3-D point positions.
- L193-197: I find this explanation a bit articulated and hard to parse. If I interpret it correctly, for each pair $(i, j)\in \mathcal{Sel}\subset M\times N$ , where $\mathcal{Sel}$ are the retained pairs (Line 192), a mean pose $\mathbf{T}\overline{ij}$ is computed from all the other pairs $\{(m, n)\in \mathcal{Sel}\ \backslash(i, j)\}$, and then the "views" that are removed at each iteration are the _pairs_ $(i^\prime, j^\prime) := \arg\max_{(m, n)\in\mathcal{Sel}} \textrm{dist}(\mathbf{T}{mn}, \mathbf{T}\overline{mn})$, where $\textrm{dist}$ is the pose distance and $\mathbf{T}{mn}$ is the relative pose obtained from pair $(m, n)$. I find confusing the concurrent use of "view" and "view pair", and also the computation of the "mean" could be rephrased to be more clear (e.g., is the mean pair-specific, as I interpreted above?)
- L222: The idea of using [CLS] tokens for category comparison is interesting. Were other options considered, or any ablations performed?
- L297: It should be explained in the text what exactly is meant by "$15^\circ$". That is, how can a single angle parametrize a distance between poses
- L333 (Supp.): The concept of "(ViT) saliency" should be introduced
- From the real-world experiments shown in the video it seems that pose estimate is not very stable across frames. Did you evaluate quantitatively the extent of the pose robustness and its impact on manipulation (e.g., would inconsistencies lead to failures in grasping the object)?

### Missing details that should be in the Supplementary
- L226: Threshold on $\theta$
- L242-243: How was the threshold for TEASER++ chosen? This intuitively might have an impact on the results
- L252-253: "effect of reducing viewpoint overlap between the image sets of reference and target objects"
- L279-280: "a subsequent K-means cluster refinement approach" (no details on the category discovery seem to be provided in the Supp.)

### Suggestions/general questions
- What are the most important factors that produce the speed-up w.r.t. to [17] mentioned, e.g., on L75? Intuitively, by having to compute correspondences w.r.t. multiple instead of a single reference view, one would think that your method should if anything be slower than [17]
- The resizing results in the object being "stretched" in some reference views. Did you evaluate if this has an effect? Did you consider using, e.g., padding, to keep the object aspect ratio fixed, and see if this has an influence on the results?

### Typos, language and bibliography
- Figure 1, caption: Extra space before "10 s"
- L41-43: Repetition of "leverage"
- L99: Extra space before "15 object"
- L122: $\mathbb{T}_{\mathcal{R}_{1:N}}$ -> $\mathbb{T}_{\mathcal{T}_{1:N}}$
- L157: Dashes should be em dashes
- L179 and 239: "Certain" -> "A selection of the"?
- L180: "show" -> "shown"
- L188: $d$ is previously (Sec. 3.1) and later (Sec. 3.5) a function of feature maps $\Phi$, but here is defined as a function of descriptor indices
- Eq. 1: $\mathbf{u}_k$ and $\mathbf{v}_k$ are here written in bold letters, whereas the correspondences are indicated in the previous lines in the text as $u_k$ as $v_k$. There is a missing step to describe how to obtain $\mathbf{u}_k, \mathbf{v}_k$ from $u_k, v_k$
- L224-225 and Eq. 2: The meaning of the symbols $F$, $G$, and $x$ should be explicitly introduced in the text
- Eq. 2: For consistency with the previous notation (see Sec. 3.2), $\phi$ should be replaced with $\Phi$ or otherwise explained in the text
- Eq. 2: The brackets around the summation could be removed. Otherwise, I suggest adapting them so as to better fit their content ("\left(" and "\right)")
- L257-258: "not only serves [...], but also have" -> "not only serves [...], but also has" or "serves not only [...], but also _to_ have"
- L258: Unnecessary comma after "time"
- L259: "30 view" -> "30-view"
- L301: "work;" -> "work."
- L334: References [10], [27] and [39] are lacking their venue
- L339: "theWild" -> "the Wild"
- L341 (Supplementary): Missing brackets around "Section 4.3"

**Quality Of The Limitations Section:**

Limitations are addressed clearly

**Reviewer Expertise:**

4: The reviewer is confident but not absolutely certain that the evaluation is correct

**Robotics Focus:**

Sufficient demonstration on hardware

**Strengths And Weaknesses:**

The motivations and the research gaps addressed by the proposed paper are stated clearly. A detailed review of the most relevant related work is provided, with a sensible and targeted classification of the previous approaches.

The paper is generally very well written, with virtually no typos or notation inconsistencies. With very limited exceptions (mentioned in "Issues"), the language is clear and the explanations are thorough, making the paper easy to read.

The quality of the presentation is very high. Despite minor possible improvements, I find the visualizations professional and very informative, particularly Figure 1, which provides an effective overview of the method, and Figure 2, which illustrates clearly the process of finding correspondences and highlights interesting, not obvious points, such as the correlation between view-to-view similarities and true geodesic errors. The attention to the details is also evident in the video (e.g., the visualization of the actual reference-target views in the correspondence matrix, or the method-overview figure edited to visualize the moving robot arm).

A number of minor technical details, particularly in the choice of the hyperparameters, could be provided for completeness. Furthermore, for multiple explanations and additional results, though all supporting non-major points, the reader is referred to the supplementary material, but the latter does not include the aforementioned additions.

The main point that could be seen as a weakness is that the viewpoint estimation process is heavily based on the previous work of [17] (as repeatedly acknowledged), and that particularly one of the main differences, i.e, the use of multiple reference views as opposed of a single reference view (as in [17]) could render the setup more complicated. However, as shown in the experiments, this choice results in a large improvement of the pose estimation accuracy. Furthermore:

- A number of other relevant insights (e.g., the impact of feature dimensionality) or modifications (consensus filtering vs. RANSAC, cosine similarity vs L2 distance) are presented
- The method runs in real time and was demonstrated on a real robotic system
- The whole application in a continual category learning scenario for new category discovery is novel and of high relevance
- Several interesting ideas are introduced or investigated, for instance the use of the [CLS] token for category comparison, the evaluation of feature similarity in 3-D rather than in pixel space, or the use of the attention maps to produce a segmentation mask

**Summary Of Recommendation:**

The paper presents a number of novel insights on the use of vision transformers for one-shot category level object pose estimation and novel category discovery, which could significantly question the need for training object- or category-specific feature networks. The approach runs in real time and is demonstrated on a robotic platform. The explored setup, i.e., category-level, without the assumption of CAD models, instance-level networks, or pose-labeled data, and only relying on a small number of reference and target views, has a high potential of concrete use in robotics.

The experiments are solid and the presentation is professional and effective for illustrating the method and the results.

I believe this paper could be a strong contribution to this Conference, after addressing a few minor points.

---

> ### Author Response · Authors · 2022-08-27
> **Authors' response to review (1/2)**
>
> Thank you so much for your time in reviewing our paper, and we are delighted to receive such detailed and insightful comments. We hope that we can address the issues that you have raised in your review here in the comments, and in a revised version of the manuscript. To answer the issues in order:
>
> \textit{L144: In what sense "normalised"?}: Our apologies, we’ve committed a cardinal statistics sin here: this should read “standardised”. We will update this. Prior to dimensionality reduction with PCA, we scale feature maps to have zero mean and unit variance.
>
> \textit{Sec. 3: What values were used for M and N?}. This is a good question. Table 2 shows results for 3 different settings where $M=10,20,30$ and $N=M$. It can be seen (taking the #Dim=32 examples) that these give 66.2%, 69.8% and 72.1% accuracy at 30º respectively. CO3D sequences contain up to 102 frames, so we will include results for even large frame numbers in the Appendix. In a practical setting, it may make sense to collect a diverse set of viewpoints of a reference object (large M) to maximise performance. At runtime on a robot, initially only a few views of a target object are available (smaller N). As mentioned in a reply to Reviewer DGYB, a key desideratum is that there be at least one reference-target image pair with meaningful correspondences between them.
>
> \textit{"Three consensus filters"....where does the "three" come from?}. Good question! This was chosen empirically. Three iterations gives best performance in our experiments which simply use view retrieval ($\textbf{RC}$ in Table 1). NB: There was a data entry error in the paper under submission for \textbf{R} results, and a slight improvement to hyperparameters throughout. A new Table 1 is attached, showing that $\textbf{RC}$ outperforms $\textbf{R}$ by a considerable margin (e.g. 26.7% better Acc30).
>
> In the attached Table 1, you can see that $\textbf{UC}$ performs no better than $\textbf{U}$. The pure retrieval settings ($\textbf{R}, \textbf{RC}$) benefit more from the consensus operations than does our full method with Umeyama’s algorithm and RANSAC. Intuitively, running Umeyama’s algorithm with RANSAC provides an alternative mechanism for consensus. Nonetheless, consensus is advantageous here, providing a ~2% accuracy improvement. $\textbf{RC}$ does not require depth, and so we include them not just as an ablation but also as a potential alternative to the full method (and one which recovers only orientation, not translation) in the depth-free setting.
>
> Use of “good” in Fig 2b: By this we mean views with high similarity (for each target, the reference view with highest similarity is chosen) and with potentially further consensus filtering to find view pairs that represent similar relative pose estimates. We’ll tighten the language here.
>
> Choice of K: We use K=50 in all our experiments. We found by ablation that reducing this can have a slight negative effect on pose estimation. In general, the method doesn’t appear to be too sensitive to this. We will add an ablation in any camera ready version.
>
> Regarding feature map (28x28) vs image (224x224) resolution - a good question. We inpaint depth maps at 224x224, then downsize to 28x28 with a nearest neighbours approach (akin to OpenCV’s “INTER_NEAREST”. This resizing mode prevents background and foreground depths being interpolated to produce a meaningless depth for patches on object edges. Retaining RGB-D resolution in the subsequent process would be a desirable improvement, but in the opinion of the authors would require a method of producing a higher resolution feature map. Amir et al’s “Deep ViT Features as Dense Visual Descriptors” run a 28x28 patch ViT with overlapping patches to achieve a 56x56 feature map without further training. However, computational bottlenecks in the quadratically scaling self-attention operation quickly kick in.
>
> L193-197: Thank you for describing this process better than we did! Your understanding is exactly right, and we will take inspiration from your notation to describe this more clearly. ‘View pairs’ should be used throughout: assuming that a certain reference view is identical to a certain target view induces a pose estimate between the sequences. It is against the mean of estimates that we filter out views.
>
> Regarding [CLS] tokens for category comparison. Yes, we tried other feature representations, including mean patch descriptors, and nearest neighbours matching to patch descriptor cluster centres. We didn’t include this in the supplementary, but will do so. As noted in a reply to Reviewer kBkt, the [CLS] token is key, also, to our approach to segmentation.

---

> ### Author Response · Authors · 2022-08-27
> **Authors' response to review (2/2)**
>
> How can a single angle parametrize a distance between poses? This distance is the geodesic error on SO3, defined between two rotation matrices $R_{1}, R_{2}$ as $d_{\mathrm{geo}}\left(R_{1}, R_{2}\right)=\cos ^{-1}\left(\frac{\operatorname{trace}\left(R_{1}^{T} R_{2}\right)-1}{2}\right)$. Intuitively, imagine that a rotation between the two poses was parameterised with an axis-angle representation: the geodesic error is the size of this angle.
>
> Thank you for pointing out that ViT saliency (L333 Supp.) is not introduced. We defined saliency as the attention scores between every patch and the [CLS] token: intuitively, this is how much a patch informs the overall semantic feature for an image, and is thus a very good metric for segmentation.
>
> Regarding pose stability in real-world experiments. This is a fair challenge, and a similar point was raised by Reviewer DGYB and addressed in our response. We did not leverage our pose estimation directly for manipulation, as category-level grasping requires more than just 6D pose (e.g. [10, 11], which use keypoints relevant to a successful grasp). Assuming an object is grasped (and there are many object-agnostic grasping approaches), we believe our pose estimates would be sufficiently stable for object placement in most cases. However, there is definitely an accuracy trade-off when approaching pose estimation in this challenging one-shot setting.
>
> We are really grateful for your diligence in drawing our attention to some missing parameters and definitions in the supplementary. We will ensure that these are all included in the final work. Thank you also for your immensely helpful list of typos - we have taken all of these in. We chose the noise threshold on TEASER++ to be the same as the inlier threshold on Euclidean distance used in our RANSAC stage, which was 0.2 (the objects in CO3D all have longest side = $2\pi$).
>
> On the speedup compared to [17]: this is an excellent question. Our speed improvements come from implementing a fast depth-inpainting method, and CUDA parallelisation of both the descriptor cosine similarity computations, and the rigid body solution with Umeyama’s method and RANSAC. We further remove the time consuming logarithmic feature binning and K-means processes used in [17]. In [17], the depth inpainting method could take ~5 seconds (with a duration that was stochastic depending on the the amount of NaN/zero-depth pixels in an image), while in our work, fast kernel convolution based filtering inpaints depth images in 2.2ms per frame. Removing logarithmic feature binning saves 30ms per frame, and further reduces correspondence calculation time, as we use at most 384-D descriptors (while log-binning produces 6,528-D descriptors). Removing the K-means clustering step that [17] uses to encourage diversity amongst the found correspondences saves considerable time: 1.2s per frame-to-frame comparison with 6528-D features, and 230ms per frame with 384-D descriptors. Overall, a 1-vs-10 comparison with [17] would take approximately 20 seconds, while a 10-vs-10 comparison in our method runs at 15Hz (66ms/frame), including time spent waiting for sensor images. Some of these insights are mentioned (e.g. Sec 7.1.3 in Appendix), but we will add a designated section to the appendix detailing the contributions to speedup over [17] (and will post in these comments).
>
> Finally, regarding ‘stretching’ of the object under the resizing (to 224x224 for the ViT). We did try padding, but were not surprised to find this gave poorer results. DINO ViT is trained on image crops with no padding but with ImageNet images that are resized to 224x224. Thus, the original dataset contains stretched images, so we believe that stretching is more within distribution than padding is.
>
> Again, thank you so much for your review, and we hope that these comments address some of your questions. Our apologies that during the rebuttal period we have not found time to upload all of these changes, but we are working on these and believe that they would significantly improve a camera-ready version of the paper.

---

### Official Review · Reviewer_DGYB · 2022-07-20

**Originality:** Good
**Technical Quality:** Good
**Clarity Of Presentation:** Good
**Impact:** 3

**Recommendation:**

Weak Reject: I recommend rejecting the paper, but will not argue for my recommendation if the majority of other reviewers have a different opinion.

**Summary:**


This paper proposes an approach for one-shot category-level pose estimation. Given M reference views of an object belonging to a specific category (which defines a category-level pose) and N views of a novel instance of that category the task is to define the category-level pose estimate. The authors propose to use a pre-trained DINO ViT to produce 2D-2D semantic correspondences between the reference and target images and then solve for the relative pose by (1)  lifting the 2D correspondences to 3D using provided depth images and (2) aligning the correspondence sets using RANSAC.

**Issues:**

List issues to be addressed during the period of author response and revision. The authors have the opportunity to submit a revised manuscript during the period. (max 5000 characters)

* Can the authors expand on how category-level pose is actually defined in the extension of the CO3D dataset. It would be useful to have this in the appendix at least so the reader doesn’t need to refer to [17].
* The authors should include metrics on the translation errors of the pose estimates after orientation. Translation is just as important as orientation in terms of usefulness in a downstream robotic application.
* How would the method perform in a more cluttered setting that is typical of robotics scenarios? Would be helpful to have some quantitative and/or qualitative results here.
* Can you expand on why you get such a speedup compared to [17], the underlying algorithms seem to be the same with exception of the current work using multiple views.
* The authors should discuss how the method would handle more cluttered scenes (as is typically encountered in robotics scenarios). Would any significant changes be needed to the approach.

**Quality Of The Limitations Section:**

Additional details required

**Reviewer Expertise:**

4: The reviewer is confident but not absolutely certain that the evaluation is correct

**Robotics Focus:**

Sufficient demonstration on hardware

**Strengths And Weaknesses:**

The paper is an extension of prior work [17] (note that this work is only on arxiv and I was unable to find any conference/journal publication of [17]). In particular the main innovation of this work seems to be in using multiple views $$I_{R_{1:M}}$$ of the reference object instead of just one view as in [17].


**Strengths**


The main strength of the paper is that it tackles an interesting and challenging problem of one-shot category-level pose estimation. They demonstrate that using multiple reference views leads to significantly improved performance over prior work [17].


**Weaknesses**



* The main experiments (Table 1, Table 2) assume 10+ views of the target object at inference time. This is typically not a realistic assumption in most robotics settings and thus requires more justification if this is to be the main test condition of the proposed approach. How would this affect the method in terms of the consensus and retrieval steps?
* The robot experiments are quite weak in the current paper. More qualitative examples of the pose estimates would be helpful. From the video it doesn’t seem that the pose estimates would be accurate enough to actually enable any meaningful robot manipulation tasks (e.g. category-level tasks such as those performed in [10, 11], category-level grasping, category-level pick and place, etc.)

* The authors should include at least some comparison the performance of this method vs. what can be achieved with a supervised category-level pose estimator such as [6, 8]. This would allow the reader to understand the tradeoffs between using a supervised approach (that has more requirements/restrictions) vs. a more general approach such as proposed one. A comparison on a standard category-level pose estimation benchmark such as the NOCS dataset from [6] would make sense.

**Summary Of Recommendation:**

Overall this paper tackles the interesting and challenging problem of category-level pose estimation without category-level training, only relying on multiple reference views of an example from the category. The current approach leverages a pre-trained DINO ViT model to extract correspondences which are then used in a fairly standard RANSAC + Umeyama algorithm for producing pose estimates. The current paper lacks comparisons to some supervised category-level pose estimation baselines (such as [6,8]). Additionally it is not at all clear that these pose estimates are sufficiently accurate to actually be used to perform category-level tasks. The method also has several limitations. In particular it only considers a 7D pose transform (rigid + scale) for characterizing the category, which may fail to capture the intra-category shape variation in certain categories (e.g. comparing a sneaker to a tall boot).

---

> ### Author Response · Authors · 2022-08-22
> **Authors' response to review (1/2)**
>
> Thank you very much for taking the time to write a detailed review of our paper, and for the points that you have raised. We hope to address some of these in the following comments, and would welcome further discussion before the close of the rebuttal period.
>
> Thank you for pointing out that a thorough explanation of how category-level 6D pose is defined in the extension of the CO3D dataset would be useful in the appendix. We will add this. In brief, CO3D consists of 51 categories, with approximately 380 video sequences of ~80 frames each for each category. The pose-labelled subset from [17] selects 20 categories, and annotates the relative 6D pose (orientation, translation) between point clouds for 10 sequences drawn from each category. As camera extrinsics for each frame relative to the point clouds are known, this induces 6D relative pose between any pair of images drawn from these labelled sequences.
>
> Thank you also for suggesting that translation errors be included - we will add a table to the appendix containing these, and will post it in our comments here soon.
>
> On the speedup compared to [17]: this is an excellent question. Our speed improvements come from implementing a fast depth-inpainting method, and CUDA parallelisation of both the descriptor cosine similarity computations, and the rigid body solution with Umeyama’s method and RANSAC. We further remove the time consuming logarithmic feature binning and K-means processes used in [17]. In [17], the depth inpainting method could take ~5 seconds (with a duration that was stochastic depending on the the amount of NaN/zero-depth pixels in an image), while in our work, fast kernel convolution based filtering inpaints depth images in 2.2ms per frame. Removing logarithmic feature binning saves 30ms per frame, and further reduces correspondence calculation time, as we use at most 384-D descriptors (while log-binning produces 6,528-D descriptors). Removing the K-means clustering step that [17] uses to encourage diversity amongst the found correspondences saves considerable time: 1.2s per frame-to-frame comparison with 6528-D features, and 230ms per frame with 384-D descriptors. Overall, a 1-vs-10 comparison with [17] would take approximately 20 seconds, while a 10-vs-10 comparison in our method runs at 15Hz (66ms/frame), including time spent waiting for sensor images. Some of these insights are mentioned (e.g. Sec 7.1.3 in Appendix), but we will add a designated section to the appendix detailing the contributions to speedup over [17] (and will post in these comments).
>
> Regarding your comments on the limitations of this method under extreme intra-category shape variation, we agree - and we identify this as a key limitation, discussed on lines 297-301. Several interesting works (e.g. Tian et al’s 2020 ECCV paper “Shape Prior Deformation for Categorical 6D Object Pose and Size Estimation”, which extends NOCS) attempt to leverage explicit or learnt parameterisations/shape priors for a given category. It remains unclear how this could be done in a zero-shot or one-shot setting, however. Nonetheless, we share your enthusiasm for addressing this limitation in future work!
>
> Regarding clutter, we did not directly address this in this work. However, we think this is an interesting extension to our setting, and would like to share our current thoughts on this with you. The DINO ViT that drives the correspondences we use has been shown to have features amenable to zero-shot semantic segmentation (e.g. Amir et al’s “Deep ViT Features as Dense Visual Descriptors”, or Melas-Kyriazi et al’s “Deep Spectral Methods”). In our work, we segment the object from background using a threshold on patch attention to the [CLS] token (a measure we call salience). In a multi-object setting, these works show that feature clustering can separate different objects. Given semantic segmentation, assignment from objects in the scene to reference categories would follow our continual learning framework. Occlusions may detract from performance, but provided meaningful correspondences can be found, we believe our method would still produce acceptable 6D pose estimates in the cluttered setting.

---

> > ### Comment · Reviewer_DGYB · 2022-08-26
> > **Response to authors rebuttal (1/2)**
> >
> > >Thank you for pointing out that a thorough explanation of how category-level 6D pose is defined in the extension of the CO3D dataset would be useful in the appendix. We will add this. In brief, CO3D consists of 51 categories, with approximately 380 video sequences of ~80 frames each for each category. The pose-labelled subset from [17] selects 20 categories, and annotates the relative 6D pose (orientation, translation) between point clouds for 10 sequences drawn from each category. As camera extrinsics for each frame relative to the point clouds are known, this induces 6D relative pose between any pair of images drawn from these labelled sequences.
> >
> > I still don't understand your definition of category-level pose. What is the 7D transform that aligns a [sneaker](https://www.google.com/aclk?sa=l&ai=DChcSEwig07-sjOX5AhVtFdQBHYVPCNYYABAHGgJvYQ&sig=AOD64_23IPBFUoPczx8VbwNButMkslac1A&adurl&ctype=5&ved=2ahUKEwi4x62sjOX5AhXtg2oFHeVjDvsQvhd6BQgBEIcB) to a [high-heel boot](https://www.amazon.com/Eldof-Womens-Knee-High-Heeled-Stiletto/dp/B07XL8K3JC)? This is not well-defined in my opinion. Alternate representations such as keypoints [10], NDFS [11] or NUNOCS [A] would seem to handle these intra-class shape variations in a more principled manner.
> >
> > [A] Wen, Bowen, et al. "Catgrasp: Learning category-level task-relevant grasping in clutter from simulation." 2022 International Conference on Robotics and Automation (ICRA). IEEE, 2022.
> >
> >
> > Thank you for your explanation of the speedups compared to [17].

---

> > > ### Author Response · Authors · 2022-08-27
> > > **Response to review's comment on category-level pose definition**
> > >
> > > Thank you for your response here, and we understand your point a better now. As we said in our original response, the assumption that category-level pose can be entirely captured by a 6D (or 7D, including scaling) transform is definitely over restrictive in some cases. However, category-level 6D pose is a firmly established setting, and many datasets reflect the same conventions as the pose-labelled CO3D dataset. For example, ObjectNet3D, Pascal3D, Objectron and NOCS, the main category-level pose datasets known to these authors, capture category-level pose with a 6D or 7D transformation. We note that this can be overly restrictive in our limitations section, and your example of the alignment of sneaker to high-heeled boot is a good one for demonstrating where this assumption breaks down.
> > >
> > > NUNOCS relaxed this somewhat by incorporating separate scaling for each axis (9D transform), but would still struggle with the example you gave.
> > >
> > > Keypoints [10] and NDFS [11] do not give single object pose estimates, so we believe that they tackle a different problem. Keypoints are very well suited to grasping, while pose can be used in object-based SLAM, object placement or goal representations for RL. It is true that category-level keypoints may be a more 'faithful' way of capturing category-level spatial relationships, but these authors believe there is still utility for category-level pose, especially in the very low data setting handled in this work.
> > >
> > > Thank you for following up on this, as we agree that this is a very interesting open problem in robotics. The difference in downstream use of keypoint-based object understanding vs pose estimate-based representations is something we believe is still being understood in robotics.

---

> ### Author Response · Authors · 2022-08-22
> **Authors' response to review (2/2)**
>
> We find the comment that multiple (10+) views of a target object is typically not realistic in robotics to be an interesting one. In the opinion of the authors of this paper, one of the key capabilities afforded to embodied agents (but not to other vision systems) is the ability to explore their worlds. Even for a robot without a mobile base, such as a manipulator, there is almost always scope to view a scene, or object in a scene, from multiple angles. It is true that in fully supervised or single-instance settings, pose estimation (e.g. Wang et al’s original NOCS paper) or grasping (e.g. Sundermeyer et al’s ICRA paper “Contact-GraspNet”) can be done from a single RGB-D image, and in this sense you are right to suggest that our method may make greater demands on multiple views than other robotics work. However, we feel that this should be viewed as a noteworthy tradeoff: in exchange for using large supervised training datasets (our method works from a handful of views of a reference object), we instead impose what we believe is a reasonable test-time burden, that of collecting several views of the target object. It should also be noted that our ‘requirement’ for multiple views of a target object serves to increase the chance that, between the reference and target views, there will be image pairs containing semantic correspondences. A limitation of using correspondences – even zero-shot ones – to drive pose estimation is that valid correspondences between the reference and target images must exist. However, this need for some correspondences between views is common to many 3D vision applications, such as VO or SfM. As shown in our attached videos, our method can begin making pose estimates from the moment of receiving the first RGB-D image of a target object, but estimates are likely to improve as further views are collected.
>
> To summarise our points on this matter, multiple views are not a hard requirement, but our method enables us to take advantage of them when they are available, and in general this leads to significantly better empirical performance. Without additional training data, two views with no correspondences give no information about relative 6D pose.
>
> Regarding your comments on the robot experiments (“it doesn’t seem that the pose estimates would be accurate enough to actually enable any meaningful robot manipulation tasks…such as those performed in [10, 11]”) we would like to assert that - while we don’t necessarily share this conclusion - the transfer of grasps between objects within a category is not in any case the focus of this work, and category-level 6D pose estimation does not generally give rise to grasp transfer. The latter (with, as you say, [10,11] being good examples) almost universally use keypoint-based specification of grasps, with [10] using hand-crafted costs on selected keypoints. Our work solves the separate problem of 6D object pose estimation, and we make no claims about grasp transfer in the paper. We know of no literature on 6D pose estimation that uses this directly to transfer category-level grasping - though we would welcome suggestions - and so we don’t feel that it would be reasonable to expect a 6D pose estimation method to enable category-level grasp generalisation.
>
> Finally, we would like to address your comments on comparing to a supervised method. We would have liked to have run our method on NOCS, but we unable to find camera extrinsics in the dataset (this has been noted before on Github: https://github.com/hughw19/NOCS_CVPR2019/issues/52), meaning we could not apply our method here. We make no claims that our one-shot method would work better than a supervised method, and we hope that (as with much zero-shot or one-shot work) it is implied that with sufficient training data in a supervised you can likely outperform these results.
>
> We are very grateful to you for the time and care that you have taken to review our work, and your comments have been very helpful to us. We look forward to any further discussions.

---

> > ### Comment · Reviewer_DGYB · 2022-08-26
> > **Response to authors**
> >
> > >We find the comment that multiple (10+) views of a target object is typically not realistic in robotics to be an interesting one. In the opinion of the authors of this paper, one of the key capabilities afforded to embodied agents (but not to other vision systems) is the ability to explore their worlds.
> >
> > I agree that an embodied agent can collect multiple views. However the diversity of these viewpoints will be limited in most practical applications (e.g. observing a shoe sitting on a shoe rack, grabbing a mug from a shelf). In these settings you will likely only be viewing the object from a fairly small set of orientations. The authors make a fair point that there is a tradeoff in terms of accuracy vs. number of views and I agree with this.
> >
> > >Regarding your comments on the robot experiments (“it doesn’t seem that the pose estimates would be accurate enough to actually enable any meaningful robot manipulation tasks…such as those performed in [10, 11]”) we would like to assert that - while we don’t necessarily share this conclusion - the transfer of grasps between objects within a category is not in any case the focus of this work, and category-level 6D pose estimation does not generally give rise to grasp transfer. The latter (with, as you say, [10,11] being good examples) almost universally use keypoint-based specification of grasps, with [10] using hand-crafted costs on selected keypoints. Our work solves the separate problem of 6D object pose estimation, and we make no claims about grasp transfer in the paper. We know of no literature on 6D pose estimation that uses this directly to transfer category-level grasping - though we would welcome suggestions - and so we don’t feel that it would be reasonable to expect a 6D pose estimation method to enable category-level grasp generalisation.
> >
> > My point is that given that the this is a robotics conference (and a robotics focused paper) the authors should demonstrate (or at least discuss) on some task for which their proposed pose estimation framework is actually useful. Right now there is no robotic application discussed in the paper. [10,11,A, B] are examples of using category-level representations for accomplishing some actual robotic tasks. [A] tackles grasping, while [10,11,B] complete tasks that involving grasping and placing. The authors should discuss (or ideally provide experiments, although I understand that this isn't possible within the rebuttal timeframe) of how this perceptual representation can be used in a robotic application. It doesn't need to be grasping as in [A], but it should be some relevant real-world task. Otherwise we don't have a sense of whether this representation is useful. Having real robot experiments in the final submission would clarify whether the accuracy of the pose estimates are sufficient.
> >
> >
> > [A] Wen, Bowen, et al. "Catgrasp: Learning category-level task-relevant grasping in clutter from simulation." 2022 International Conference on Robotics and Automation (ICRA). IEEE, 2022.
> > [B] Wen, Bowen, et al. "You Only Demonstrate Once: Category-Level Manipulation from Single Visual Demonstration." arXiv preprint arXiv:2201.12716 (2022).
> >
> >
> > > Finally, we would like to address your comments on comparing to a supervised method. We would have liked to have run our method on NOCS, but we unable to find camera extrinsics in the dataset (this has been noted before on Github: https://github.com/hughw19/NOCS_CVPR2019/issues/52), meaning we could not apply our method here. We make no claims that our one-shot method would work better than a supervised method, and we hope that (as with much zero-shot or one-shot work) it is implied that with sufficient training data in a supervised you can likely outperform these results.
> >
> > Fair enough, I didn't know that NOCS was missing camera intrinsics/extrinsics information. However I think that some sort of quantitative comparison to another learning based method method (not an ablation of your own method) would be helpful.
> >
> >
> > Additionally you could consider using your Dino-VIT based correspondences inside the Teaser++ framework (instead of using RANSAC) as TEASER++ is robust to outliers. This could alleviate the need for doing viewpoint pruning as described in 3.4.1

---

> > > ### Author Response · Authors · 2022-08-27
> > > **Response to reviewer's comment**
> > >
> > > We agree that experiments with applied examples of how pose estimation would be used might strengthen the work, though we hope that as presented, the paper already presents a capability that is often viewed as standalone in robotics conferences (that is, many other 6D pose papers at robotics conferences do not demonstrate physical interaction). As you note, there hasn't been time in this rebuttal period to run manipulation experiments, but we are currently putting together a system that allows object placement according to our pose estimates, and hope that we could include qualitative results in a camera ready version of this work. We agree that grounding the pose estimation ability in downstream robotics tasks would be a meaningful contribution to the work.
> > >
> > > Thank you for the suggestion of using our correspondences within TEASER++. This is a good idea, and we will try this!

---

> ### Comment · Reviewer_DGYB · 2022-08-26
> **Updated Review**
>
> Overall the concerns I expressed in my original review still remain and I think that the paper could be improved by addressing these issues. That being said, after thinking it over I believe that the paper offers an interesting investigation of how to use pre-trained backbone features (Dino-VIT in this case) for use in a category-level pose estimation task. Thus I am **leaning towards updating my recommendation to a weak accept**.

---

> ### Author Response · Authors · 2022-08-27
> **Translation results**
>
> Attached is a table showing results for the translation estimation component of 6D pose, for the four versions of our method presented in Table 1 in the original paper.

---

### Official Review · Reviewer_kWPG · 2022-08-01

**Originality:** Good
**Technical Quality:** Good
**Clarity Of Presentation:** Very Good
**Impact:** 3

**Recommendation:**

Weak Accept: I recommend accepting the paper, but will not argue for my recommendation if the majority of other reviewers have a different opinion.

**Summary:**

This paper presents an approach to object pose estimation at the category level by leveraging multiple views of a single reference object from the category. The approach combines advantages of different paradigms : deep models and feature matching.  High dimensional features from a pre-trained ViT is used for efficient descriptors through dimensionality reduction. These set of image descriptors from multiple views of an object are used to find correspondences between the target and the reference object of the same category. An average pose estimate is obtained by filtering best reference-target view pairs. Finally the filtered subset are aligned using a robust least square estimation to give a 6D object pose.
The paper builds upon [17] by using multiple reference object views which leads to considerable improvement in performance.

**Issues:**

Comparative evaluations mentioned earlier.

**Quality Of The Limitations Section:**

Limitations are addressed clearly

**Reviewer Expertise:**

4: The reviewer is confident but not absolutely certain that the evaluation is correct

**Robotics Focus:**

Sufficient demonstration on hardware

**Strengths And Weaknesses:**

Strengths:

The paper is well written and supported with relevant media content.
The technical contribution is incremental yet considerable.
The proposed approach results in better performance in terms of speed and accuracy than existing category level technique.

Weakness:
Although the performnace improvement in Table 1 has promising results, the extensive evaluations with other techniques would be desirable.
For e.g., [14], You Only Demonstrate Once: Category-Level Manipulation from Single Visual Demonstration- RSS2022
Also, in line 38, I do not think its well argued.  Do you mean method in [14] is designed for single object instances? Please check

**Summary Of Recommendation:**

I believe the approach is interesting and results in improved performance than recent approach.
However, I do want to see more comparative evaluations to be convinced.

---

> ### Author Response · Authors · 2022-08-22
> **Authors' response to review**
>
> Thank you for taking the time to review our work, and for your succinct and accurate summarisation. We are pleased that you have noted the much better performance (in terms of both speed and accuracy) that our method gives versus the baselines in this challenging pose estimation setting.
>
> Regarding your point around line 38, we will work to make this clearer in future versions of the manuscript. We currently say “These approaches…are designed for single object instances and thus fail to exhibit desirable category-level generalisation”, and we believe this to be correct, but it is possible that our phrasing is not clear. To clarify, the works about which this has been said ([13,14,15]), perform pose estimation for particular objects (which we refer to as single object instances), with respect to some reference view(s) for those exact objects. What we mean to highlight with this line is that this is in contrast to our method, in which the reference object may not be exactly the same as the test time object, but can instead be a different instance from the same category. We hope this explanation clarifies this somewhat.
> Thank you for the reference of Wen et al, ‘You Only Demonstrate Once: Category-Level Manipulation from Single Visual Demonstration’ [RSS2022]. We don’t believe this would be an appropriate comparison as it would not be possible to use this method in our setting. This is because for Wen et al to generalise to novel instances from an object category, they have to train on a very large set of CAD models of different objects from that category. The crucial contribution of our work is that it generalises one-shot, leveraging just a few views of a reference object to produce a reference frame for any object from the same category. This is a challenging setting, and therefore aside from [17] and TEASER++, we were unable to find any further suitable baselines. We would welcome further suggestions, however.
>
> Thank you again for taking time to review our work, and we look forward to responding to any further comments or concerns you may have.

---

### Official Review · Reviewer_kBkt · 2022-08-03

**Originality:** Good
**Technical Quality:** Good
**Clarity Of Presentation:** Good
**Impact:** 3

**Recommendation:**

Weak Accept: I recommend accepting the paper, but will not argue for my recommendation if the majority of other reviewers have a different opinion.

**Summary:**

This work tackles the task of 6D pose estimation for novel objects given prior observations of a single object of the same category (where multiple RGB-D images from known relative poses are available for both, the template and the query object, and the goal is to find a rotation and translation to align the template and the query object).

The proposed approach is to: a) extract spatial features from all images using a pre-trained ViT (with dimensionality reduction using PCA) , b) find a rotation that aligns the template and query object using a set of (query image, template image) pairs that match well and are consistent with other pairs, and c) estimates a 7D alignment (R, t, scale) using patchwise matches between the selected pairs and leveraging RANSAC.


**Issues:**

- Please see the Weakness section. In particular, I would be interested in the author response to why directly reasoning in 3D space and simply searching for (robustly via RANSAC) a 7D pose that maximizes alignment between matching 3D points is not a good solution.
- As a minor concern, the only evaluation in the paper focuses on rotation estimation, and it would be good to see some results on the accuracy of the full pose estimated.

**Quality Of The Limitations Section:**

Limitations are addressed clearly

**Reviewer Expertise:**

4: The reviewer is confident but not absolutely certain that the evaluation is correct

**Robotics Focus:**

Highly relevant to robotics but no hardware experiments

**Strengths And Weaknesses:**

Strengths:
- The paper tackles a relevant problem and considers a natural setup for it. Given multiple views of a query object and a template object, computing alignment between the two is a great formulation for the task of one-shot pose estimation..

- The experiments are reported on the Co3D dataset which represents a challenging setup, and shows that the proposed method does indeed generalize across objects.

- A key implementation detail that this work highlights is the benefit of feature dimensionality reduction. This is well-ablated and leads to consistent improvements.

Weaknesses:

- One concern is that the paper relies rather heavily on the framework presented in [17], and the primary contribution here is to extend that framework to leverage multiple query views (in a slightly straightforward manner). While this obviously leads to significant empirical gains (as using multiple views should!), this does limit the technical contribution. Moreover, this should also necessitate an alternate baseline where one simply uses the approach from [17] to get multiple estimates and chooses the ‘mode’.

- While the work does use multiple query views, the precise methodology feels somewhat constrained. In particular, the only patch ‘correspondences’ that are allowed (and ultimately drive the optimization) are between batches at the same grid locations across images. This is suboptimal for two reasons as: a) it implicitly assumes that images from very similar viewpoints are available for matching and may not be a valid assumption, and b) cannot leverage observations of matching content from dissimilar viewpoints (the handle of a cup might match between a left-view and a right-view, and should drive the pose estimation).

- On a note slightly related to the above, I am curious why a more ‘traditional’ SfM like approach is not considered (as the method, or atleast as a baseline). In particular, given depth images, the patches can be considered as 3D points with associated (deep) features. One could then construct matches between the 3D points observed for the source and target objects, and using RANSAC, optimize the 7D pose to best align them. I would be interested to hear why this would not suffice (particularly as it is the ‘typical’ approach that drives large-scale structure from motion — and in this case the depth + known extrinsics within each object would only simplify the task further)?

- (comparatively minor concern) The continual learning aspect seems a bit weak and is more of a post-hoc application rather than something integral to the method. In particular:
a) the notion of a ‘category’ does not really keep evolving: if one sees multiple chairs, the category is still represented by a single template (or perhaps a collection of independent template — the text is unclear on this). In any case, multiple examples of a category do not lead to an evolving notion of the category.
b) The determination of a ‘new’ vs seen category is simply done via average cosine similarity of the [CLS] token of the ViT model and does not have much in common with the approach for 6D pose alignment.
Considering both these, the continual learning aspect of the work, while tackling a good goal, seems a bit adhoc and technically disconnected.


**Summary Of Recommendation:**

Overall, I am slightly torn about this work. On the one hand, it considers a relevant setup and shows promising results. However, this is currently outweighed by the concerns (similarity to [17], and possibilities of a simpler SfM-like solution).

Update:
Thanks for the response, and in particular for addressing my misunderstanding regarding the matching, which also addressed the concerns regarding a more 'regular' SfM-like method. I am happy to update the rating towards an accept and would recommend acceptance as a poster.

---

> ### Author Response · Authors · 2022-08-22
> **Authors' response to review**
>
> Thank you for taking the time to write a detailed review of this paper. We are very pleased to see that you share our enthusiasm for this unconstrained pose estimation setting and believe it to be an important problem for robotics. We would like to address some of the points you raised in your review, and hope that we can provide some helpful clarification.
>
> Regarding the relationship between this work and [17], we feel that it is important to note that while our method leverages the concept of DINO-ViT based correspondences from [17], the extension to multiple views in a real-time formulation provides a fundamentally different capability, and produces a solution that is – unlike [17] – of significant practical utility, especially in a robotics setting. While the method in [17] takes, on average, 10 seconds to run a single frame (against 10 ‘reference’ frames), our method is capable of running up to 10-against-10 frames at 15Hz, and achieves much improved accuracy (for the CO3D dataset, an improvement from 49.4% accuracy at 30º for [17] to 69.8º in our method, an increase of over 20%). Our speed improvements come from implementing a fast depth-inpainting method, and CUDA parallelisation of both the descriptor cosine similarity computations, and the rigid body solution with Umeyama’s method and RANSAC.
>
> We believe that your suggestion that results for full pose estimation, rather than just orientation, be presented is a good one, and we will be updating the paper with results for 3D position prediction (translation), and will post the table in a comment, too. Together with orientation, these capture the full 6D pose estimates. Other combined metrics used in the literature, such as ADD, are not suitable for cross-instance pose estimation as they assume the shapes aligned are identical.
>
> Regarding the finding of patch correspondences, we believe that you may have misunderstood our approach . We certainly do not constrain these correspondences to only be between patches in the same grid locations. As you note, this would be an extremely restrictive assumption, as it would require that objects had identical shapes, and were imaged from identical viewpoints. As described in Sec 3.2, for each image pair we find a set of P correspondences. A patch in Image 1 may form a correspondence with a patch from any grid location in Image 2. To re-emphasise this point, when finding correspondences between two 28x28 feature maps (a total of 28*28=784 patches per feature map), we compute $784^2$ cosine similarities. Thanks to our fast CUDA implementation, this – even when scaled to comparing many-against-many images – takes only a few milliseconds (see Tab. 2 in the main text). We hope this helps to clarify our approach, and we will ensure that this is made more explicit in subsequent versions of the manuscript.
>
> In light of the above clarification, we believe that your suggestion about using 3D points derived from the correspondences to compute a 7D transform estimate, with RANSAC, may have been based on the same misunderstanding, as this is indeed exactly how our method operates. As you suggest, once correspondences have been found, they are lifted into 3D given the depth images, and Umeyama’s algorithm, with RANSAC, is used to compute an optimal 7D pose between the images using least squares. This is covered in Section 3.4.2 (lines 199-206).
>
> Thank you for your comments around the approach and formulation for continual category discovery in this work. Regarding this section seeming ‘adhoc’, we would have liked to have dedicated more emphasis to this aspect of the proposed method but were constrained by space. We do believe that we present a robust strategy for this process, though, and that having a solution to this is setting greatly improves the scope for deploying the proposed pose estimation technique in real world settings. The use of the [CLS] token in the category retrieval step was chosen as the approach giving best performance (measured by average precision) in test settings constructed from the CO3D dataset. The same [CLS] token is key to our pose estimation method elsewhere, as we use the attention values between each patch and the [CLS] token as a saliency measure, a threshold on which is used for segmenting the objects in the images, to ensure that all correspondences are between objects rather than background. There is thus, in our opinion, a close relationship, both technically and functionally, between the pose estimation and continual learning parts of our method.
>
> Finally, your review notes that there are no hardware experiments, but these are described in Section 4.3, as well as the supplementary and video. The video contains 3 examples of our pose estimation method being run in real time (15Hz) with a Panda robot arm with a wrist-mounted RealSense camera.
>
> Thank you again for taking time to review our work, and we look forward to responding to any further comments or concerns you may have.

---

> > ### Comment · Reviewer_kBkt · 2022-08-27
> > **Updated Review**
> >
> > Thanks for the response, and in particular for addressing my misunderstanding regarding the matching, which also addressed the concerns regarding a more 'regular' SfM-like method. I am happy to update the rating towards an accept.

---

> ### Author Response · Authors · 2022-08-27
> **Translation results**
>
> Attached is a table showing results for translation estimation for the 4 versions of our method presented in Table 1 in the original paper.

---

### Meta-Review · Area_Chair_r5iZ · 2022-08-11

**Recommendation:** Accept (Poster)
**Confidence:** 4

**Metareview:**

Reviewers felt positively that the proposed task is important and challenging and the proposed method is reasonable.  The reviewers also felt that the results are very nice:
- The dataset used for evaluation is challenging
- The performance is better than baselines in terms of speed and accuracy
- The results show nice generalization
- Nice ablations on dimensionality reduction
- The method runs in real-time
- The paper includes real-robot experiments
Reviewers also felt that the paper is well-written and visualizations are clear and informative. Reviewers appreciated the response in the rebuttal and the subsequent improvements to the paper.

At the same time, reviewers felt that the paper had some weaknesses, especially a lack of comparison to more baselines (such as a supervised category-level pose estimator [6, 8, 14]).  Although the setting is different (one-shot vs training on a full dataset), it would still be helpful to show the gap in performance between such methods to get a sense of how far the performance of the proposed method is compared to methods that can train on more data. Other useful baselines include TEASER++ or a baseline that uses NOCS or NUNOCS for category-level generalization.

It would also be helpful to show the performance as the number of views at test-time is varied below 10. Reviewers also requested to see more qualitative examples of the pose estimates. Finally, reviewers would like to see a more clear explanation of how the ground-truth pose 7D pose is defined across instances of a category.

The paper would be greatly improved by including these modifications in the final version of the paper.


**Best Paper Nomination:**

No